# Adolescent Distress: Is There a Vaccine? Social and Cultural Considerations during the COVID-19 Pandemic

**DOI:** 10.3390/ijerph20031819

**Published:** 2023-01-19

**Authors:** Francesco Demaria, Stefano Vicari

**Affiliations:** 1Child and Adolescent Neuropsychiatry Unit, Bambino Gesù Children’s Hospital, IRCCS, Viale Ferdinando Baldelli 41, 00146 Rome, Italy; 2Department of Life Sciences and Public Health, Catholic University, 00168 Rome, Italy

**Keywords:** adolescent distress, vaccine, social and cultural changes, COVID-19 pandemic

## Abstract

The COVID-19 pandemic had an unprecedented impact on mental health. In particular, the impact on adolescents was likely significant due to vulnerability factors linked to this developmental stage and pre-existing conditions of hardship. The present work aimed at grasping the particular effects of the pandemic on social and cultural aspects of adolescence, providing a cross-sectional picture of this historical moment of contemporary youth culture. Further research is needed to verify the findings.

## 1. Introduction

The COVID-19 pandemic had an unprecedented impact on global health and mental health [1,2]. In particular, the impact on adolescents may have been significant [3] due to vulnerability factors linked to their developmental stage and level of education as well as pre-existing mental health conditions [4].

During the pandemic, adolescents were forced to change their daily routines in order to adapt to new health and social rules, including the use of distance learning, the limitation of social contact, and the suspension of sports activities, with negative effects for their psychophysical health [5,6]. Additionally, fear of contagion, the threat to family integrity, and experiences of mourning, isolation, and loneliness in addition to constant media bombardment [7] may have contributed unfavorably to adolescents’ mental health. Racine et al. [8] estimated that, globally, in the first year of the pandemic, 25% of children and adolescents suffered from depression, and 20% suffered from anxiety—rates that doubled those of the pre-pandemic era. Furthermore, Raymond et al. [9] found symptoms of post-traumatic stress and anxiety in young people with socio-emotional vulnerability during the pandemic, noting that girls and adolescents exhibited the most symptoms compared to boys and children.

During the pandemic, psychiatric urgency also increased exponentially: an analysis by the Italian Society of Pediatrics found a worrying increase in emergency room attendance in nine Italian regions. Specifically, an 84% increase during the period March 2020–2021 was noted in comparison to the prior year (i.e., pre-COVID). The pathologies showing the greatest increase in emergency room access were suicidal ideation (+147%), depression (+115%), and eating disorders (+78.4%) [10]. United Nations International Children’s Emergency Fund (UNICEF) data (2021) reveal that, globally, suicide is the fifth-leading cause of death among youths aged 10–19 years, and the risk of suicide increases with age [11]. Similarly, in Eastern Europe and Central Asia, suicide is the leading cause of death among adolescents aged 15–19 years.

During the delicate and exceptional period of the COVID-19 pandemic, mental health services could not guarantee the necessary support [12]. However, this alone does not justify the data showing profound unease among young people [13,14]. Historical crises such as the COVID-19 pandemic can mark an entire generation [15,16]. In addition to causing psychological stress and even trauma, such highly negative events [17] may also have social implications with significant consequences for young subjects. For adolescents, it has been shown that the COVID-19 pandemic profoundly affected different domains of life [18], impacting the ways in which they managed their social relationships; balanced their needs of autonomy versus attachment [19]; engaged in educational, learning, and professional endeavors; solved problems; defined and evaluated the surrounding world; and formed perspectives and an optimistic vision of the future [20]. The pandemic called all of these tasks into question and exacerbated pre-existing critical issues [21,22]. Moreover, COVID-19-related distress resulted in vulnerabilities such as increased social isolation [22,23], difficulties with online learning [24], and mental health problems [25,26,27]. Wider social and cultural considerations of these phenomena are needed.

## 2. Research Objective

The new culture of communication that was triggered by the pandemic underlines the need for a new concept of adolescence as a lived moment, with associated beliefs and realities, rather than merely a transitional phase of development. The scientific literature describes the widespread effects of the pandemic, efforts to manage these effects, and the pathological consequences of the pandemic on youths. However, it does not sufficiently explore the current reality of youths, who are not only fragile but also the constituents of future society. This appears to be a current and central societal problem.

The work carried out so far has investigated critical issues related to the resolution of the pandemic (e.g., vaccinations) or psychological distress and mental health problems in selected groups. The academic research of these works, with verification of the sources, was carried out using the electronic databases *PubMed*, *CINAHL*, *PsycInfo*, *MedLine*, and *Cochrane Library* by inserting the keywords of this article (Table 1)

The present work aimed at grasping the particular effects of the pandemic on social and cultural aspects of adolescence, providing a cross-sectional picture of this historical moment of contemporary youth culture. 

The paper aims at eliciting a better understanding and a new concept of adolescence as not merely an important transition phase in development but also a prolonged period of life characterized by individual traits and new (i.e., digital) patterns of communication. These characteristics may expose adolescents to additional discomfort and suffering.

## 3. Let Us Stop the Pandemic of Adolescents: Social and Cultural Considerations

Adolescence is a developmental period characterized by transition and transformation, which places significant stress on psychic balance. During this stage, new aspirations, desires, and emotions accompany a maturation and development of the body, modifying the relationship between the adolescent and their surrounding world. Indeed, adolescence is characterized by not only physical development and biological growth but also major changes in social roles. 

Studies on adolescence originated in Europe and the United States at the beginning of the twentieth century. The founding father of scientific research on adolescence was the American psychologist and pedagogist Granville Stanley Hall [53], who described adolescence as a turbulent and problematic phase of development connected with biological changes. Later, the anthropologist Margaret Mead in 1928 [54] demonstrated, in her study of young people in the Samoan Islands, that adolescent changes are a culturally determined “product”. Sigmund Freud in 1936 [55] then contributed the psychoanalytic perspective, underlining the importance of drive development in puberty. (S. Freud stated that the first recapitulation of childhood sexuality (i.e., the first moment at which a “summary” of prior events is taken to help understand the present) occurs during adolescence. This represents a critical task in which drive development plays a significant role. The second recapitulation occurs in the climacteric.) Subsequent models of adolescence included the cognitive model of Jean Piaget [56], linked with the development of hypothetical-deductive thought (determining a speculative/introspective attitude) in adolescence, and the psycho-social model of Erik Erikson in 1950 [57], which broadened Freud’s psychosexual development framework and placed it in a social framework, considering adolescence one of several phases of the “life span”—the outcome of which is determined by crisis and conflict. (E. Erikson’s model was based on the concept of the identity crisis. He introduced the idea that the search for identity fully manifests in adolescence and proposed an evolutionary scheme characterized by eight stages—each of which corresponds to a psychosocial crisis that, if overcome, represents a step towards psychological maturity.) Psychologists, sociologists, and anthropologists have therefore studied adolescence for a long time. However, there is no clear and linear discourse capable of integrating the different perspectives of the various disciplines. We can thus assume that adolescence is influenced by biological, psychological, cultural, and social factors.

Over the last century, the timing of these processes has changed. Puberty, which is the psychophysical process that initiates the adolescent period, now occurs at a younger age in almost all populations, while role changes (i.e., completion of education, marriage, parenthood) have been delayed. Thus, today, the transition from childhood to adulthood takes longer [58]. During this extended period, profound social and cultural changes may have a significant impact on adolescents’ health and well-being [59,60]. Although three-quarters of mental disorders onset prior to the age of 25 years, adolescents have rarely been the subject of mental health research and interventions [61]. Adolescents are fragile, vulnerable, and experience-seeking, and these characteristics may increase their risk of discomfort, dependence, and deviance if they do not find adequate support from their family members and educational environments. Manifestations such as social withdrawal, self-harm, technology addiction, toxicophilic behavior, dis-control/disruptive behavior, deviant behavior, anxiety, depression, and eating disorders are psychopathological expressions of this discomfort, and the boundaries of these conditions are often labile and easily confused [62]. Cultural context and ethnicity, in addition to religion and stigma, may also play a fundamental role in determining discomfort and access to treatment across cultures [63].

During adolescence, the definition of one’s identity must attune with the changes and growth of one’s body. Physical transformations are particularly prevalent in early adolescence, while psychic and mental maturation are more evident in later adolescence. At a neurobiological level, the cerebral system is involved in the maturation process between adolescence and young adulthood through the structural alteration of both white and gray matter. The limbic area (responsible for emotional reactions, behavioral responses, short- and long-term memory, and learning) matures earlier than the prefrontal cortex (responsible for executive functions, planning, and impulse control). The result of this misbalance is often reduced inhibition accompanied by increased perceptive intensity and a tendency to seek new experiences. Additionally, sex hormones, which increase until reproductive capacity is reached, become more active, particularly in the limbic system, amplifying the search for new and exciting experiences [64,65,66]. 

This so-called “sensation-seeking” [67] drive, characterized by a continuous search for new stimuli and sensations, can lead adolescents to engage in risky behaviors. This neurobiological substrate is the basis of all growth and maturation during adolescence and also in terms of the attempt to define one’s identity.

Physical and psychological changes during adolescence may be associated with feelings of uncertainty and instability. Adolescence is a delicate transition phase characterized by physiological and natural anxiety and depression in response to growth and change. Adolescents require adequate coping mechanisms to navigate these changes—and their identity formation process—successfully.

The growing adolescent unease witnessed in recent years has no doubt been shaped by the social and cultural reality, which has determined behavioral models that are poorly adapted to physiological and psychological identity construction. As a result, adolescents may be more susceptible to behavioral drifts and emotional disturbances that, in some cases, may trigger angry, dark, and alienating aggression.

Given the tasks of experimentation and identity formation that characterize adolescence, social and cultural changes may have an outsized impact on adolescents [68]. In particular, the COVID-19 pandemic may have triggered an acute cultural discomfort among adolescents [21]. The maturational process during adolescence calls individuals to experiment, put themselves to the test, develop autonomy, and, above all, learn how to accept mistakes, suffer, and grow accustomed to experiences of “psychic pain”. However, in the current environment, adolescents are less called to put their own resources on the line. As a result, they are less likely to experience “healthy reactions” and, above all, less likely to experience “pain” as a stimulus for change and growth. Modern society promotes a performative reality in which failure is not allowed [69]. Suffering is less tolerable, and the achievement of (personal) goals is paramount. Thus, the elimination or inoculation of obstacles and the achievement of control are of the utmost importance.

During adolescence, parental communication (within the family) has been shown to have the strongest effect on the prevention of discomfort that on the deviance [70]. However, a communicative parental relationship (in contrast to that in which parents’ main role is to educate and protect children) is less relevant today, when many adolescents are seeking to develop their autonomy against the background of parental incapacity [71,72]. Thus, it seems that paternal authority (based on an authoritarian transmission of limits and rules) and parenthood (marking generational differences) has receded in the face of family socialization based on parent–child leveling, with rights and duties negotiated continuously. In this vein, many families are now formed under a pressure of team spirit, seeking to establish a group of equals (i.e., parents and children) that neglects generational ties and boundaries as well as traditional parental responsibilities towards children [73].

The period of adolescence is characterized by a normative “identity crisis” predicated on physical and moral growth that leads the young person to feel that they have outgrown their childhood aspirations and must begin to search for an adult identity [73]. As a result, the young person begins to detach from parental figures and create their own social network. They may develop a secret life (inaccessible to parents) constituted by friendships, love affairs, entertainment, a diary or blog, and social networks. This transition can be complex.

During adolescence, parents lose influence, while the influence of peers becomes greater. Some friendships may involve peer authority, whereby a peer leader becomes an identifying figure replacing paternal or parental authority. In this context, adults are no longer models, and instead, adolescents model adults. Such modeling may make adolescents feel safe and allow them to recognize themselves more easily. They may use the same language as adults but fail to perceive limits and recognize hierarchies. In this process, adolescents may turn to technology and digital communities to nourish their inner circle rather than develop into adulthood [74]. To be themselves, they have to sufficiently differentiate themselves from others while establishing enough similarity to others to avoid being mocked.

In the absence of social orientation (i.e., in the context of individualization), the adolescent seeks a personal identity, sometimes reproducing characters they identify with and drawing on the surrounding environment (e.g., consumer culture). Today, beauty is an object of mass culture and is fueled by marketing, social networks, magazines, and the “cult of youth”, among other factors. Accustomed to online forums and messaging, the younger generation is likely to have multiple virtual identities. Not feeling any state of mind in the face of artifice, they may be led to transform their appearance readily in an attempt to aestheticize their presence in the world [75]. The construction of identity during adolescence also depends on the ability to integrate one’s “new” body into a self-image according to societal norms. Body image is a fundamental factor that contributes to adolescents’ overall well-being [76]. At the same time, it can increase adolescents’ experiences of vulnerability [77]. Adolescents’ increasingly frequent gender dysphoria represents an incomplete construction of identity based on a lack of bodily acceptance.

The skin and the brain constitute two fundamental biological parts with a common origin in the ectoderm [78]. The adolescent’s changing skin prompts them to accept their new body and new identity. Moreover, the skin represents the place of contact, which may be open or closed to the world according to affective circumstances. It is therefore a shock absorber of tensions from both the outside and the inside [79].

It is no coincidence that one of the most common behaviors in adolescence is non-suicidal self-harm, such as cutting, scratching, and burning [80,81]. Self-harm is considered an act of aggression. It is particularly common among girls, who may use it to cope with their transforming body [82]. While self-harm is a deliberate act motivated by different causes, it is typically aimed at “appeasing” unbearable psychic pain. For many adolescents, psychic pain is unacceptable and must be counteracted immediately. There is no possibility of enduring it, so action is taken immediately to alleviate the pain. The shift from mental pain to physical pain makes suffering visible and thereby controllable, even if it remains unacceptable [83]. This action may become a repeated behavior, implemented automatically—and without explanation—in reaction to suffering. Adolescents in this cycle will inflict pain on themselves in an effort to “drive out” their suffering.

Other risk behaviors in relation to the body have similar social connotations. Girls are more likely than boys to assume discreet, silent forms of bodily suffering, including eating disorders, scarification, and suicide attempts. Boys, on the other hand, are more likely to demonstrate violent, challenging, and risky behaviors [73].

In contemporary society, body image has found its maximum expression through technology [84]. Adolescents are spending an increasing amount of time online, communicating with each other. In line with this, the possibilities of connection offered by mobile devices and social media have reached unprecedented levels. Thus, digital devices have become powerfully attractive to young people seeking affiliation, social approval, and novelty.

Lachance [69] discussed the singular relationship that “hypermodern teenagers” have with temporality through their significant use of new technologies, which has determined a new culture of communication. In both traditional and contemporary societies, rituals and myths structure time. Through these, children inherit a “significant relationship with temporality”. For adolescents, digital technologies and cultural products modify and induce a new relationship with time. In a society with uncertain contours, where true independence always comes later, temporality maybe seized by young people in an attempt to emancipate themselves.

Concerns have been raised that the constant connectivity offered by the internet can cause serious mental health problems among adolescents. Therefore, understanding the effects of the digital culture may allow us to better understand adolescent behaviors that may express discomfort or addiction. To date, research has produced few relevant data on the potential relationship between digital exposure and mental health problems in adolescents. In a longitudinal study, Jensen M et al. [85] found no increase in mental health symptoms in adolescents who spent more time using technology. Teens at the highest risk of mental health problems also showed no additional symptoms on days with greater use of technology. Furthermore, Odgers CL and Jensen M [86] explored the link between the use of digital technology and adolescent mental health, with a specific focus on depression and anxiety. Their review found that most prior research has been correlational, focused on adults (versus adolescents), and generated few relevant—and often conflicting—data. Even the most recent and rigorous large-scale studies considered in the review have reported only small associations between levels of daily digital technology use and adolescent well-being without characterizing a specific significance. Masciantonio et al. [87] examined the relationships between well-being and the active or passive use of various social networks (e.g., Facebook, Instagram, Twitter, TikTok) during the COVID-19 pandemic. Passive Facebook use was related to social comparison, which, in turn, was associated with lower well-being. Active Instagram use was related to social support, which, in turn, was associated with higher life satisfaction but also greater negative affect. Regarding Twitter, active usage was related to social support, which, in turn, was associated with higher life satisfaction; however, passive use was negatively associated with social comparison, which, in turn, was associated with greater negative affect. In contrast, use of TikTok showed no relation to well-being.

Discussions conducted on social networking platforms (e.g., pro- or anti-COVID-19 vaccine fora, engagement with social media influencers) may reveal mass behavior related to important and delicate political and health issues [33,88]. Improved knowledge of this digital culture may help us to investigate trends and opinions, user cohesion, and the spread of misinformation on social media and other platforms [89]. In turn, this may help public health agencies eliminate misinformation and thereby enhance vaccine uptake [90] and also increase positive messages in order to educate people and reduce the negative consequences of the COVID-19 pandemic [91].

Digital development has changed our ways of communicating, relating, and “being in the world”. The ability to make quick contact and establish immediate communication as well as the publication of images online have created a new social culture. However, this new culture of connection cannot replace lived relationships [92]. Social networks allow users to “show” themselves (in images) and perpetually renew their character in an endless game while waiting for comments that may reconstruct the meaning of the depicted moments [73]. Walter Benjamin [93] distinguishes between “life” and “experience”, defining experience as a lived experience that has been the subject of subjective work. Since youth culture is characterized by a multiplication of actions in a very short space of time, young people must develop strategies to transform their experiences into meaningful lived experiences. It is therefore not surprising that, during the COVID-19 pandemic, young people were most affected by the effects of the lockdown and other restrictions as well as the direct use of virtual channels. Vall-Roqué et al. [94] found that the impact of the COVID-19 lockdown on the use of social networks (e.g., Instagram, YouTube, TikTok, Twitter, Facebook) was associated with body image disturbances and low self-esteem as well as a tendency to lose weight and a greater risk of eating disorders among adolescent girls and young women. In particular, the pandemic-related impacts on mental health in adolescents manifested in increased depression, anxiety, and suicidal ideation [95]. Identifying the activities that could be promoted and those that should be limited to safeguard the emotional and behavioral well-being of adolescents in the use of social networks is foundational [96].

The interruption of school attendance during the pandemic not only negatively impacted learning and knowledge but also characterized a loss of experience, social comparison, and growth [24]. The school closures also intensified the sense of loneliness that is frequently experienced in adolescence—a developmental period that prioritizes inclusion and peer acceptance [18]. It is not the intent of this discussion to criticize the protection measures adopted to defend the community from COVID-19 but to reflect on adolescent needs and requirements in order to improve the support that is offered to this community during exceptional times [34].

The COVID-19 pandemic represented one of the most profound challenges to the educational system, forcing schools to re-assert their role in spreading knowledge and making children and youth aware of their choices and actions in support of their personal growth. The pandemic may also lead us to improve our understanding of adolescent distress amidst recent social and cultural changes. A broader and more inclusive definition of adolescence is needed to define laws, social policies, and service systems appropriate to this developmental stage. Current shortcomings in this respect must be counteracted by the dissemination of good practices that can lead to the sustained, efficient, and equitable delivery of mental health care [97].

## 4. Conclusions

The COVID-19 pandemic had a significant impact on adolescent mental health. Recent social and cultural changes have also had an unprecedented impact on this population. In particular, changes to the role of the family, educational institutions, and digital technologies (determining different ways of managing relationships, sociality, self-perception, and body image) have led to the development of new behavioral models that are not well-adapted to the times and therefore less suited to supporting an integrated physiological and psychological construction of identity.

Digital technologies are the future and have become pervasive in the lives and relationships of both young people and adults. Policies limiting adolescents’ access to new technologies could prove harmful given that such technologies are frequently used as a source of social support [98] and are necessary for participation in the marketplace. Thus, while the unease of adolescents may be partly attributed to their use of new technologies, governments must take responsibility for managing social policies and service networks to improve the relationship between adolescents and digital technologies in order to encourage their healthy adaptation and well-being. Future data are needed to verify the claims made.

## Figures and Tables

**Table 1 ijerph-20-01819-t001:** Previous works.

Zhang P et al. [28], O Afifi T et al. [29], Wang D et al. [30], Leonhardt JM/Pezzuti T. [31], Burak D. [32], Cascini F et al. [33], Meherali S et al. [34], Sallam M. [35], Bardosh K et al. [36], Xiao Y et al. [37], Fadda M et al. [38], Moffitt TE et al. [39], Jones B et al. [40], Jung H/Albarracín D. [41], Miconi D et al. [42], Buzzi et al. [43], Kaplan AK et al. [44], de Filippis R et al. [45], Darmody M et al. [46], Anderson D et al. [47], Simkhada P et al. [48], Okabe-Miyamoto K/Lyubomirsky S. [49], Presidential Policy/Strategy Unit [50], Gittings L et al. [51], Crowley P/Hughes A. [52].

## Data Availability

Not applicable.

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
