# Peer review of "Adolescent Distress: Is There a Vaccine? Social and Cultural Considerations during the COVID-19 Pandemic"

_ijerph, 2023, doi:10.3390/ijerph20031819_

Round 1
Reviewer 1 Report
Research summary:
The topic is relevant and may be of interest to a broad range of the journal's readers. However, this reviewer has some major concerns about the paper.
Major Strengths: The major strengths of the research are:
- The topic is interesting
Major Weaknesses: The major weaknesses of the research are:
- The structure and contents of the paper need to be improved.
- The proposed discussion does not provide any useful information
Grammar and Readability:
The paper requires a detailed revision of the English and is unclear in some parts.
Specific Comments: My specific comments concerning this manuscript are:
- The abstract does not highlight the specifics of the research or findings but contains too much background information. Some details of the research would be nice for example numbers addressing the sample, data, percentage improvement, etc.. Remove some of the background material and add some details of the research. Moreover, it is good to provide some specifics (e.g., sample size, dataset size, numbers from results, etc.).
- The innovation of the paper seems limited. The proposed study is a short discussion on a widely discussed problem.
- There needs to be an explicit research objective(s) and/or research question(s) stated, preferably as a separate section. This helps readers find out what the research is trying to address.
- The goal of the paper is not clear. The paper lacks interesting content to be published in a journal.
- Related Work section is missing. Certainly, there has been more recent (within the last two years) research on this topic published in information science and/or computer science outlets. An academic search on the topic (using keywords from the manuscript’s title) shows that there is recent work in this area. I suggest some recent works start with: https://doi.org/10.1016/j.ipm.2022.103095, https://doi.org/10.1016/j.puhe.2021.11.022, https://doi.org/10.1016/j.ijid.2021.05.059
- Starting from the previous works, I suggest introducing a table to summarize the most recent works and to highlight the novelty of the proposed work.
- The reference list needs tidying up, as there are references missing items or formatting issues. Please be consistent with the formatting and use some standard formatting style.
- Please consider using a more convincing way to evaluate the proposed study. The paper lacks any scientific and analytical contributions.
- The only interesting thing is the considerations reported in Section 2. Nevertheless, the discussion is weak, and the paper lacks an experimental evaluation.
- I suggest completely revising the objective of the paper. The authors should discuss objective considerations of the problem by previously analyzing the posts published on different social networks, such as Facebook, Twitter, Tik Tok, etc.. This would make the study interesting from a scientific point of view.
Concluding Remarks:
I think that the paper could be improved with the considerations I reported in the review, but this version is not ready for publication.
Author Response
Manuscript ID: ijerph-2097787
Response to Reviewers' Comments:
We thank the reviewers for their scrutiny of the manuscript and insightful remarks, their very good feedback on our study was very encouraging. We hope to match their thoroughness and detail in our reply.
Please note function that our replies are written in italics and that any changes to the text are marked up using the “Track Changes”
Response to Reviewer 1
- The topic is relevant and may be of interest to a broad range of the journal's readers. However, this reviewer has some major concerns about the paper.
Major Strengths: The major strengths of the research are:
- The topic is interesting
Major Weaknesses: The major weaknesses of the research are:
- The structure and contents of the paper need to be improved.
- The proposed discussion does not provide any useful information
Grammar and Readability:
The paper requires a detailed revision of the English and is unclear in some parts.
- The abstract does not highlight the specifics of the research or findings but contains too much background information. Some details of the research would be nice for example numbers addressing the sample, data, percentage improvement, etc.. Remove some of the background material and add some details of the research. Moreover, it is good to provide some specifics (e.g., sample size, dataset size, numbers from results, etc.).
Response: Thanks for your request of clarification. The present work offers a unique contribution to the literature, with respect to contemporary aspects of youth culture. The COVID-19 pandemic was a historic moment, involving significant social and cultural changes. Such changes may be particularly impactful during adolescence, due to the experimentation and identity seeking behaviors that are characteristic of this developmental stage. Thus, adolescents may have been especially susceptible to pandemic-related discomfort and suffering. However, further research is needed to verify this claim.
We have corrected the text (abstract section and keyword) as follows:
“The COVID-19 pandemic had an unprecedented impact on mental health. In particular, the impact on adolescents was likely significant, due to vulnerability factors linked to this developmental stage and pre-existing conditions of hardship. The present work aimed at grasping the particular effects of the pandemic on social and cultural aspects of adolescence,providing a cross-sectional picture of this historical moment of contemporary youth culture. Further research is needed to verify the findings.
Keywords: adolescent distress, vaccine, social and cultural changes, COVID-19 pandemic.”
- The innovation of the paper seems limited. The proposed study is a short discussion on a widely discussed problem.
Response: Thanks for your request of clarification. While the youth theme has been significantly discussed in the literature,it hasnot yet been fully understood and addressed.
We have modified the text (2. Research objective section) as follows:
“The new culture of communication that was triggered by the pandemic underlines the need for a new concept of adolescence as a lived moment, with associated beliefs and realities, rather than merely a transitional phase of development. The scientific literature describes the widespread effects of the pandemic, efforts to manage these effects, and the pathological consequences of the pandemic on youths. However, it does not sufficiently explore the current reality of youths, who are not only fragile, but also the constituents of future society. This appears to be a current and central societal problem.”
- There needs to be an explicit research objective(s) and/or research question(s) stated, preferably as a separate section. This helps readers find out what the research is trying to address
Response: Thanks for your suggestion.
We have modified the text (2. Research objective section) as follows:
“The present work aimed at grasping the particular effects of the pandemic on social and cultural aspects of adolescence, providing a cross-sectional picture of this historical moment of contemporary youth culture.”
- The goal of the paper is not clear. The paper lacks interesting content to be published in a journal.
Response: Thanks for your suggestion. We have modified the text (2. Research objective section) as follows:
“The paper aims at eliciting a better understanding and a new concept of adolescence as not merely an important transition phase in development, but also a prolonged period of life characterized by individual traits and new (i.e., digital) patterns of communication. These characteristics may expose adolescents to additional discomfort and suffering.”
- Related Work section is missing. Certainly, there has been more recent (within the last two years) research on this topic published in information science and/or computer science outlets. An academic search on the topic (using keywords from the manuscript’s title) shows that there is recent work in this area. I suggest some recent works start with: https://doi.org/10.1016/j.ipm.2022.103095, https://doi.org/10.1016/j.puhe.2021.11.022, https://doi.org/10.1016/j.ijid.2021.05.059
Response: Thanks for your suggestion. We have modified the text (3. Let’s stop the pandemic ... of adolescents: social and cultural considerations) as follows:
“Discussions conducted on social networking platforms (e.g., pro or anti–COVID-19 vaccine fora, engagement with social media influencers) may reveal mass behavior related to important and delicate political and health issues [88,33].Improved knowledge of this digital culture may help us to investigate trends and opinions, user cohesion, and the spread of misinformation on social media and other platforms [89]. In turn, this may help public health agencies eliminate misinformation and there by enhance vaccine uptake [90], and also increase positive messages in order to educate people and reduce the negative consequences of the COVID-19 pandemic [91].”.
- Starting from the previous works, I suggest introducing a table to summarize the most recent works and to highlight the novelty of the proposed work.
Response: Thanks for your suggestion. We have modified the text (2. Research objective section) as follows:
“The work carried out so far has investigated critical issues related to the resolution of the pan-demic (e.g. vaccinations) or psychological distress and mental health problems in selected groups. The academic research of these works, with verification of the sources, was carried out using the electronic databases PubMed, CINAHL, PsycInfo, MedLine, and Cochrane Library by inserting the keywords of this article (Table 1)”.
Table 1
Zhang P et al. [28], O Afifi T et al. [29], Wang D et al. [30], Leonhardt JM/Pezzuti T.[31], Burak D. [32], Cascini F et al. [33], Meherali S et al. [34], Sallam M. [35], Bardosh K et al. [36], Xiao Y et al. [37], Fadda M et al. [38], Moffitt TE et al.[39], Jones B et al. [40], Jung H/Albarracín D.[41], Miconi D et al. [42], Buzzi et al. [43], Kaplan AK et al.[44], de Filippis R et al.[45], Darmody M et al. [46], Anderson D et al. [47] Simkhada P et al. [48], Okabe-Miyamoto K/Lyubomirsky S. [49], Presidential Policy/Strategy Unit [50], Gittings L et al. [51], Crowley P/Hughes A. [52]. |
- The reference list needs tidying up, as there are references missing items or formatting issues. Please be consistent with the formatting and use some standard formatting style
Response: Thanks for your suggestion. The reference list has been re-ordered and the formatting has been standardized.
- Please consider using a more convincing way to evaluate the proposed study. The paper lacks any scientific and analytical contributions
Response: Thanks to request of clarification. The article has been revised in an attempt to provide greater clarity to the contents presented, drawing on new scientific contributions.
- The only interesting thing is the considerations reported in Section 2. Nevertheless, the discussion is weak, and the paper lacks an experimental evaluation.
Response: Thanks for your suggestion. The discussion presented in Section 3 has been revised and enriched with additional scientific contributions. Future data will be needed to verify the claims made.
- I suggest completely revising the objective of the paper. The authors should discuss objective considerations of the problem by previously analyzing the posts published on different social networks, such as Facebook, Twitter, TikTok, etc.. This would make the study interesting from a scientific point of view.
Response: We specifically this better in the text (3. Let’s stop the pandemic ... of adolescents: social and cultural considerations- Line 234)
“Masciantonio et al. [87] examined the relationships between well-being and the active or passive use of various social networks(e.g., Facebook, Instagram, Twitter, TikTok) during the COVID-19 pandemic. Passive Facebook use was related to social comparison, which, in turn, was associated with lower well-being.Active Instagram use was related to social support, which, in turn, was associated with higher life satisfaction, but also greater negative affect. Regarding Twitter, active usage was related to social support, which, in turn, was associated with higher life satisfaction; however, passive use was negatively associated with social comparison, which, in turn, was associated with greaternegative affect. In contrast, use of TikTok showed no relationto well-being.”
Line 261
“Vall-Roqué et al. (2021) found that the impact of the COVID-19 lockdown on the use of social networks (e.g., Instagram, YouTube, TikTok, Twitter,Facebook) was associated with body image disturbances and low self-esteem,as well asa tendency to lose weightand a greaterrisk of eating disorders among adolescent girls and young women.”
Line 266
“Identifying the activities that could be promoted and those that should be limited to safeguard the emotional and behavioral well-being of adolescents in the use of social networks is basic [96]”
Thank you, again, for your feedback and suggestions.
We have re-evaluated the objective of the study in alignment with your comments, with particular consideration of the scientific perspective and the unique contribution of the work to the literature.

Reviewer 2 Report
The report pontentially addresses a very interesting and actual topic, but this is not adequately developed. It has an accurate and long introduction about what adolescence is but there is no original evidence or information about what happened/changed during the Covid period, although there already exists a consolidated scientific literature about this topic.
The report should be restructured with an updated literature which accomplishes the initial aim expressed in the report’s title.
In detail:
I suggest integrating the first paragraph of section 2 (from line 54 to 64) with more literature concerning the definition and delimitation of adolescence (maybe using a footnote) to give a more complete frame about the different approaches which define this lifetime. In fact different disciplines could disagree with the definition used by the authors.
From line 100 to line 153 there are some crucial concepts about the social and personal growth of adolescents, but these are not sustained by an adequate asset of references. I suggest integrating these paragraphs with the most relevant literature on this topic. Picking from other disciplines could reinforce this part (i.e. sociology, especially regarding at the construction of identity, the role of technology, etc.)
The last part of the paper must be integrated with some recent research, reports, papers, already published on this topic.
Author Response
Manuscript ID: ijerph-2097787
Response to Reviewers' Comments:
We thank the reviewers for their scrutiny of the manuscript and insightful remarks, their very good feedback on our study was very encouraging. We hope to match their thoroughness and detail in our reply.
Please note function that our replies are written in italics and that any changes to the text are marked up using the “Track Changes”
Response to Reviewer 2
- The report pontentially addresses a very interesting and actual topic, but this is not adequately developed. It has an accurate and long introduction about what adolescence is but there is no original evidence or information about what happened/changed during the Covid period, although there already exists a consolidated scientific literature about this topic. The report should be restructured with an updated literature which accomplishes the initial aim expressed in the report’s title.
Response: Thanks for your request of clarification. We have corrected the text (Introduction section) as follows:
“Historical crises such as the COVID-19 pandemic can mark an entire generation [15,16]. In addi-tion tocausing psychological stress and even trauma, such highly negative events [17]may also have social implications with significant consequences for young subjects. For adolescents, it has been shown that the COVID-19 pandemic profoundly affected different domains of life [18], im-pacting the ways in which they managed their social relationships; balanced their needs of au-tonomy versus attachment [19]; engaged in educational, learning, and professional endeavors; solved problems; defined and evaluated the surrounding world; and formed perspectives and an optimistic vision of the future [20]. The pandemic called all of these tasks into question and ex-acerbated pre-existing critical issues [21,22]. Moreover, COVID-19–related distress resulted in vulnerabilities such as increased social isolation [23,22], difficulties with online learning [24], and mental health problems [25-27].Wider social and cultural considerations of these phenome-na are needed”
- In detail: I suggest integrating the first paragraph of section 2 (from line 54 to 64) with more literature concerning the definition and delimitation of adolescence (maybe using a footnote) to give a more complete frame about the different approaches which define this lifetime. In fact different disciplines could disagree with the definition used by the authors.
Response: Thanks for your request of clarification. We have modified the text (3. Let’s stop the pandemic ... of adolescents: social and cultural considerations) using footnotes as follows:
“Studies on adolescence originated in Europe and the United States at the beginning of the twen-tieth century. The founding father of scientific research on adolescence was the American psy-chologist and pedagogist, Granville Stanley Hall [53], who described adolescence as a turbulent and problematic phase of development, connected with biological changes. Later, the anthro-pologist Margaret Mead in 1928 [54] demonstrated, in her study of young people in the Samoan Islands, that adolescent changes are a culturally determined “product”. Sigmund Freud in 1936 [55]then contributed the psychoanalytic perspective, underlining the importance of drive de-velopment in puberty1. Subsequent models of adolescence included the cognitive model of Jean Piaget [56], linked with the development of hypothetical-deductive thought (determining a speculative/introspective attitude) in adolescence, and the psycho-social model of Erik Erikson in 1950 [57], which broadened Freud’s psychosexual development framework and placed it in a social framework, considering adolescence one of several phases of the “life span”— the out-come of which is determined by crisis and conflict2. Psychologists, sociologists, and anthropolo-gists have therefore studied adolescence for a long time. However, there is no clear and linear discourse capable of integrating the different perspectives of the various disciplines. We can thus assume that adolescence is influenced by biological, psychological, cultural, and social factors.
1 S. Freud stated that the first recapitulation of childhood sexuality (i.e., the first moment at which a "summary" of prior events is taken to help understand the present) occurs during adolescence.This represents a critical task, in which drive development plays a significant role. The second recapitulation occurs in the climacteric.
2 E. Erikson's model was based on the concept of the identity crisis. He introduced the idea that the search for identity fully manifests in adolescence and proposed an evolutionary scheme characterized by eight stages—each of which corresponds to a psychosocial crisis that, if overcome, represents a step towards psychological maturity”.
- From line 100 to line 153 there are some crucial concepts about the social and personal growth of adolescents, but these are not sustained by an adequate asset of references. I suggest integrating these paragraphs with the most relevant literature on this topic. Picking from other disciplines could reinforce this part (i.e. sociology, especially regarding at the construction of identity, the role of technology, etc.
Response: Thanks for your suggestion. We integrate the paragraphs with the reference mentioned as follows:
Line 143:
Kinghorn A, Shanaube K, Toska E, Cluver L, Bekker LG. Defining adolescence: priorities from a global health perspective. Lancet Child Adolesc Health. 2018 May;2(5):e10. doi: 10.1016/S2352-4642(18)30096-8. Epub 2018 Mar 23
Line 144:
Deng, W.; Gadassi Polack, R.; Creighton, M.; Kober, H.; Joormann, J. Predicting negative and positive affect during COVID‐19: A daily diary study in youths. J Adolesc Res 2021, 31, 500-516. doi:10.1111/jora.12646
Line 150:
Lachance J. L’adolescence hypermoderne. Le nouveau rapport au temps des jeunes. Presses de l’universitè Laval, coll. < Sociologie au coin de la rue>, 2012.
Line 157:
Jensen M, George MJ, Russell MA, Lippold MA, Odgers CL. Does Adolescent Digital Technology Use Detract from the Parent-Adolescent Relationship? J Res Adolesc. 2021 Jun;31(2):469-481. doi: 10.1111/jora.12618. Epub 2021 Apr 7.
Winstone L, Mars B, Haworth CMA, Judi Kidger J. Social media use and social connectedness among adolescents in the United Kingdom: a qualitative exploration of displacement and stimulation. BMC Public Health. 2021 Sep 24;21(1):1736. doi: 10.1186/s12889-021-11802-9.
Line 162:
Le Breton D. Corps et adolescence. Editions Fabert. 2016.
Line 163:
“The period of adolescence is characterized by a normative “identity crisis,” predicated on physi-cal and moral growth that leads the young person to feel that they have outgrown their child-hood aspirations and must begin to search for an adult identity [73]. As a result, the young per-son begins to detach from parental figures and create their own social network. They may de-velop a secret life (inaccessible to parents), constituted by friendships, love affairs, entertain-ment, a diary or blog, and social networks. This transition can be complex.”
Line 175:
Dienlin T, Johannes N. The impact of digital technology use on adolescent well-being. Dialogues Clin Neurosci. 2020 Jun;22(2):135-142.doi: 10.31887/DCNS.2020.22.2/tdienlin.
Line 178:
“In the absence of social orientation (i.e., in the context of individualization), the adolescent seeks a personal identity, sometimes reproducing characters they identify with, drawing on the sur-rounding environment (e.g., consumer culture). Today, beauty is an object of mass culture, fueled by marketing, social networks, magazines, and the “cult of youth”, among other factors. Accustomed to online forums and messaging, the younger generation is likely to have multiple virtual identities. Not feeling any state of mind in the face of artifice, they may beled to trans-form their appearance readily in an attempt to aestheticize their presence in the world [75]”.
Line 186:
Jiménez Flores P, Jiménez Cruz A, Bacardi Gascón M. [Body-image dissatisfaction in children and adolescents: a systematic review]. Nutr Hosp. 2017 Mar 30;34(2):479-489. doi: 10.20960/nh.455
Line 187:
Carvalho GX, Nunes APN, Moraes CL, Veiga GVD. Body image dissatisfaction and associated factors in adolescents. Cien Saude Colet. 2020 Jul 8;25(7):2769-2782. doi: 10.1590/1413-81232020257.27452018.
Line 190:
The skin and the brain constitute two fundamental biological parts with a common origin in the ectoderm [78]. The adolescent’s changing skin prompts them to accept their new body and new identity. Moreover, the skin represents the place of contact, which may be open or closed to the world according to affective circumstances. It is therefore a shock absorber of tensions from both the outside and the inside [79].
Line 198:
Betz DE, Sabik NJ, Ramsey LR. Ideal comparisons: Body ideals harm women's body image through social comparison. Body Image. 2019 Jun;29:100-109. doi: 10.1016/j.bodyim.2019.03.004. Epub 2019 Mar 20.
Line 202:
Muehlenkamp JJ, Brausch AM. Body image as a mediator of non-suicidal self-injury in adolescents. J Adolesc. 2012 Feb;35(1):1-9. doi: 10.1016/j.adolescence.2011.06.010. Epub 2011 Jul 20.
Line 209:
Revranche M, Biscond M, Husky MM.[Investigating the relationship between social media use and body image among adolescents: A systematic review]. Encephale. 2022 Apr;48(2):206-218. doi: 10.1016/j.encep.2021.08.006. Epub 2021 Nov 18.
Line 214:
“Lachance [69] discussed the singular relationship that “hypermodern teenagers” have with tem-porality through their significant use of new technologies, which has determined a new culture of communication. In both traditional and contemporary societies, rituals and myths structure time. Through these, children inherit a “significant relationship with temporality”. For adolescents, digital technologies and cultural products modify and induce a new relationship with time. In a society with uncertain contours, where true independence always comes later, temporality maybe seized by young people, in an attempt to emancipate themselves.”
- The last part of the paper must be integrated with some recent research, reports, papers, already published on this topic.
Response: Thanks for your suggestion. We have modified the text (Line 253) as follows:
“Social networks allow users to“show” themselves (in images) and perpetually renew their char-acter in an endless game, while waiting for comments that may reconstruct the meaning of the depicted moments [73]. Walter Benjamin [93] distinguishes between “life” and “experience,” de-fining experience as a lived experience that has been the subject of subjective work. Since youth culture is characterized by a multiplication of actions in a very short space of time, young people must develop strategies to transform their experiences into meaningful lived experiences..”
Thank you, again, for your feedback and suggestions.
We have developed the work with the scientific contribution of different disciplines

Round 2
Reviewer 1 Report
Thanks to the authors for following the previous comments. The paper has been sufficiently improved and can be published.
Reviewer 2 Report
The authors have sufficiently addressed reviewer's modification requests